# Predictors of the Third (Booster) Dose of COVID-19 Vaccine Intention among the Healthcare Workers in Saudi Arabia: An Online Cross-Sectional Survey

**DOI:** 10.3390/vaccines10070987

**Published:** 2022-06-21

**Authors:** Sami Alobaidi, Almoutaz Hashim

**Affiliations:** Department of Internal Medicine, University of Jeddah, Jeddah 21959, Saudi Arabia; ahahashem@uj.edu.sa

**Keywords:** predictors, COVID-19 vaccine, booster dose, healthcare workers, KSA

## Abstract

**Objective:** The COVID-19 pandemic is still continuing throughout the world, with newer genetic variants regularly appearing from different parts of the world. Considering the waning of immunity against COVID-19 infection even with two doses of the COVID-19 vaccine, regulatory authorities have authorised booster COVID-19 vaccination in many countries, especially for vulnerable populations, including healthcare workers. The current study analysed factors predicting the third (booster) dose of COVID-19 vaccine intention, including the health belief model (HBM), among the healthcare workers in Saudi Arabia. **Methods:** The current study was a cross-sectional online survey performed from 1st October 2021 to 30th November 2021, using a questionnaire prepared in Google^TM^ form among healthcare workers in Saudi Arabia. The questionnaire asked demographic factors, COVID-19 experience of participants, subjective assessment of health, intention of COVID-19 booster dose vaccination, preferences for local/foreign-made vaccines, and health belief of the study population related to COVID-19 infection and COVID-19 booster dose. **Results:** This study received 2059 complete responses. The study population reported mixed health belief with respect to the susceptibility of COVID-19 infection, and higher health belief perception regarding the severity. The perceptions of the study participants regarding the benefits of COVID-19 booster dose were positive. There were few barriers to COVID-19 booster dose expressed by study participants. A total of 1464 (71.1%) study participants reported positive intent for receiving a COVID-19 booster dose. The study showed significant association between definite intention to receive a booster dose and nationality (*p* = 0.001), marital status (*p* = 0.017), gender (*p* < 0.001), education level (*p* = 0.001), monthly income (*p* < 0.001), and co-morbid medical illness (*p* = 0.045). The perception of the COVID-19 booster vaccine as a good idea to minimise worries about getting COVID-19 (OR = 2.28, CI 1.89–2.76), and perceptions that receiving the third (booster) dose reduces the risk of COVID-19 infection and associated complications (OR= 2.69, CI = 2.17–3.34), of the perceived benefit construct of HBM, predicted significantly higher definite intention to receive a booster dose. The concern with the safety of the vaccine (OR= 0.40, CI 0.34–0.47) under the perceived barriers construct of HBM predicted as significantly higher no definite intention to receive a booster dose. **Conclusions:** The results of the present study can guide policy makers in their efforts to promote booster doses of COVID-19 vaccination among the healthcare workers in Saudi Arabia.

## 1. Introduction

Since the World Health Organization (WHO) declared the novel coronavirus (COVID-19) as a pandemic on 11 March 2020, its spread continues throughout the world, with newer genetic variants regularly appearing from different parts of the world [1]. The most important available option in the control of the pandemic at present is vaccination against COVID-19 infection [2]. Currently, available evidence has clearly established the effectiveness of available vaccines against the novel coronavirus in reducing the severity of the COVID-19 infection and associated mortality [3]. Unfortunately, emerging evidence from observational data also indicates that the ability of the vaccines in protecting against COVID-19 infection diminishes progressively over time, and more significantly among older adults [4,5]. Considering the waning of immunity against COVID-19 infection after usual doses of COVID-19 vaccine, the debate was going on regarding introducing booster doses of vaccination among the most vulnerable populations to restore vaccine effectiveness [6]. Moreover, the available evidence also shows that the COVID-19 vaccination booster helps to achieve or improve the peak antibody levels following the primary immunisation series [7]. The detection of the new variant, Omicron (B.1.1.529), in November 2021, and its rapid spread in many countries across the world, accelerated the debate over booster COVID-19 vaccination [8]. The recent data shared by the UK Health Security Agency clearly showed that the protection against symptomatic COVID-19 infection by the Omicron variant could be less than 10% at 25 weeks after receiving regular doses of the vaccine [8]. Considering these observations, regulatory authorities have authorised booster COVID-19 vaccination in many countries, and free COVID-19 booster doses were provided by government agencies to vulnerable populations, including healthcare workers [6]. The Kingdom of Saudi Arabia (KSA) also announced the availability of booster COVID-19 vaccine doses for high-risk individuals including healthcare workers on 20 October 2021.

Since the beginning of the pandemic, KSA had put very active steps in controlling the pandemic. Soon after the release of the results of the Phase III clinical trial with the Pfizer-BioNTech COVID-19 vaccine, it was approved by the Saudi authorities for use [9]. The ministry of health also contacted healthcare workers (HCWs) directly through various means and encouraged them to voluntarily enrol for vaccine uptake [9]. Despite the efforts of government agencies across the world to vaccinate a maximum number of the population, studies showed significant hesitancy in receiving COVID-19 vaccine in various sections of the population [10]. COVID-19 vaccine hesitancy was shown to be variable among different sections of the Saudi population. A recent study from Saudi Arabia after the first COVID-19 vaccine was introduced by the government showed positive response to COVID-19 vaccine intent from 71.9% of the study subjects [10]. Another large study among 23,582 HCWs in the KSA also showed that only 64.9% showed a positive intent for COVID-19 vaccine, with males, Christians, and Pakistanis showing significantly higher positive intention [11]. Another study reported that only 50.52% of HCWs in the KSA reported positive intention to receive the COVID-19 vaccine [12]. When compared to vaccine acceptance among HCWs from other countries, the vaccine acceptance rates among HCWs from the KSA were significantly lesser. Moreover, the rate of reported acceptance was not in consistence with the actual behaviour of HCWs in vaccination uptake exemplified by a recent study among healthcare workers during the first month of its availability in the KSA, which showed that 66.73% of HCWs had not enrolled in the vaccination program [13]. Female HCWs, those who accessed vaccine information mainly from social media, and participants younger than 40 years old showed a lower level of vaccine uptake [13]. Previous studies among HCWs in the KSA regarding vaccination uptake for H1N1 and the seasonal influenza vaccination also showed significant vaccine hesitancy [14,15].

Vaccine hesitancy for a COVID-19 vaccine booster is a public health challenge. Morbidity and mortality associated with the spread of the Omicron variant can be reduced by an increased acceptance of the COVID-19 booster dose strategy, especially among the high-risk population including HCWs. However, emerging evidence suggests that vaccine hesitancy for a COVID-19 vaccine booster dose is present among a considerable proportion of the fully vaccinated general public [16]. A recent study from the Western world showed that 4% of the respondents were unwilling to receiving a COVID-19 booster dose [17]. COVID-19 booster dose vaccination hesitancy was significantly higher among healthy adults below 45 years of age, those with low levels of fear regarding catching COVID-19 infection, those not following COVID-19 preventive measures, those with lower educational attainment, and those belonging to a lower socioeconomic status [17]. There are no studies specifically exploring vaccine hesitancy for a COVID-19 vaccine booster dose among HCWs in Saudi Arabia to date. Considering the relatively higher prevalence of vaccination hesitancy among HCWs residing in the KSA, there is a need to explore the attitude towards COVID-19 vaccine booster doses among this high-risk population, in order to understand the need for specific public health interventions targeting HCWs, to improve the uptake of the COVID-19 vaccine booster dose. The aim of the current study was to explore perceptions of the booster dose of COVID-19 vaccine among HCWs from the KSA and the health belief predictors of intention to receive the booster dose of the vaccine. 

## 2. Methods

### 2.1. Study Design and Participants

The present study was an online survey using the snowball technique to recruit participants using Google^TM^ forms. This study was conducted from 1 October 2021 to 30 November 2021. The Saudi government announced the availability of the COVID-19 booster dose for high-risk individuals including healthcare workers on 20 October 2021, and the COVID-19 booster dose became available for everyone above the age of 18 on 7 November 2021. The institutional ethical committee has approved this study with approval number UJ-REC-027. The STROBE guidelines for cross-sectional studies were followed while reporting this study’s findings [18].

The questionnaire was prepared in English along with its Arabic translation. The translation of the questionnaire was performed according to the standard procedure involving independent bilingual experts. The questionnaire was pilot tested among 20 participants before the beginning of the study. The questionnaire’s link was shared with the author’s social media contacts in the health sector (questionnaire in Appendix A). Participation in this study was voluntary and online consent was obtained from each participant. It was expected from every participant to forward the study questionnaire to at least five other healthcare workers in their contact list. Only healthcare workers working in Saudi Arabia who were above 18 years of age were considered for this study.

### 2.2. Instruments

The questionnaire had four parts. The first part collected socio-demographic variables. The following personal details of the study participants were collected: age, marital status, education, gender, nationality, monthly income, occupation, and habitat. The second part of the questionnaire collected COVID-19-related variables and health self-assessment. The questions included in part 2 were regarding history of COVID-19 illness of the participants or in any close family members of the participants. A subjective assessment of health status on a 5-point scale from very good to very poor was also included in the second part of the questionnaire. The third part of the questionnaire was related to COVID-19 vaccination. The questions explored intention to receive the COVID-19 booster dose and COVID-19 vaccine preferences (local/foreign). The intention to receive a COVID-19 booster dose was rated using a four-point scale (definitely no, probably no, probably yes, definitely yes). The vaccination preference was also rated using a four-point scale (completely confident to completely non-confident).

The fourth part of the questionnaire was related to the health belief model (HBM) in the context of the COVID-19 illness and booster vaccination [19]. HBM constructs were used to explore beliefs of the study participants related to COVID-19 infection (susceptibility and severity constructs), perceptions regarding COVID-19 booster vaccine (benefits and barrier constructs), and beliefs regarding cues that might facilitate booster dose uptake (cues to action construct). Agreement with each item of the questionnaire was rated using a four-point scale (strongly agree, agree, disagree, and strongly disagree). The HBM constructs are shown in Figure 1.

### 2.3. Statistical Analysis

The Statistical Package for Social Sciences (SPSS Inc., Chicago, IL, USA, version 26.0 for Windows) software was employed in data analysis. The characteristics of the study population were explored using descriptive statistics. The statistical significance among study variables was analysed using chi-square tests as well as independent sample *t*-test. Binary logistic regression using the vaccination intention as the dependent variable was utilised to analyse the HBM construct predictors of COVID-19 booster dose uptake. The model fit of logistic regression analysis was tested using the Hosmer–Lemeshow goodness-of-fit test. The forward LR method of regression was used to obtain odds ratios (OR) and 95% confidence intervals (95% CI). The statistical significance level was kept at *p* < 0.05.

## 3. Results

### 3.1. Demographics

This study received 2059 complete responses. Table 1 shows the demographics of the study participants. The mean age of the study population was 32.92 ± 8.33 years (range 19 to 90 years). Of all study participants, 50.3% were females, 84.7% were Saudi citizens, 52.2% were married, and 69.67% had a bachelor’s degree. The study participants included doctors (37.91%), dentists (5.93%), nurses (12.26%), pharmacists (7.37%), technicians (16.16%), and others (20.37%). The majority of the study respondents were from the western province of the Kingdom (48.3%), followed by central (29%), eastern (10.7%), southern (7.9%), and northern (4%) provinces, and 37.9% of the study participants were earning > SAR 15,000 per month. A total of 18.6% reported smoking habits and 44.7% reported a sedentary lifestyle; 43.7% of the study population were directly managing COVID-19 infected patients, 23.6% reported COVID-19 infection in the past, and 59% also had a family history of COVID-19 infection. Only 16.4% of the study population had other medical co-morbidities, and 59.7% rated their subjective health status as very good. 

### 3.2. The Health Beliefs of the Study Population

Health beliefs about the susceptibility for COVID-19 infection were mixed for the study participants. Of all participants, 64.2% disagreed with the statement regarding the greater chance of getting infected with COVID-19, and only 40.2% reported their worries regarding the higher chance of getting COVID-19. However, 74.7% of survey participants still thought that acquiring COVID-19 is a possibility for them right now. The study participants had relatively higher health belief perceptions regarding the severity of the COVID-19 infection. Of all participants, 84.2% agreed that the COVID-19-related complications are serious, and 57% still admitted their fear of getting COVID-19 infection. However, only 28.3% agreed that getting COVID-19 will make them severely sick. The benefits of COVID-19 booster dose vaccination were viewed positively by the trial participants. Of the participants, 54.7% agreed with the statement that the booster COVID-19 vaccination is a good step to reduce their fear of catching COVID-19, and 69.3% agreed that receiving the booster COVID-19 vaccine decreases their chances of catching COVID-19 infection and its complications. The study participants expressed fewer perceived barriers to COVID-19 booster dose vaccination. A total of 46.1% were concerned with the possible side effects, 43.1% were concerned with the efficacy of the COVID-19 booster dose vaccination, 37.9% expressed concerns regarding the safety of the COVID-19 booster dose vaccination, and 34.2% expressed concern regarding faulty/fake COVID-19 booster dose vaccination. Regarding cues to action, 75.4% of the study population reported that they would take the booster dose of COVID-19 vaccine only if they were given adequate information about the vaccine, and 62% reported that they would only take it provided the booster dose was taken by many healthcare workers in Saudi Arabia. 

### 3.3. The Confidence in Local and Foreign-Made COVID-19 Vaccines and Preferences

A majority of the study participants (64.5%) expressed their confidence in the domestically made COVID-19 vaccine. However, a higher proportion (80.6%) reported their confidence in the foreign-made/imported vaccines. Of the participants, 47.6% expressed a preference for foreign-made COVID-19 vaccines, whereas only 14.2% expressed a preference for domestically made vaccine. There was a statistically significant link between vaccination preference and confidence in local vaccines (698.99, df 6, *p* = 0.001), with more people choosing locally manufactured vaccines. There was also a statistically significant association between vaccine preference and confidence in the foreign-made vaccines (95.36, df 6, *p* < 0.001), with a higher number of respondents who reported confidence in foreign-made vaccines preferring foreign-made vaccines. Males (26.60, df 2, *p* < 0.001), non-Saudi nationals (22.65, df = 2, *p* < 0.001), education level masters and above (41.09, df = 4, *p* < 0.001), monthly income more than SAR 15,000 (43.15, df 6, *p* < 0.001), and those with family history of COVID-19 infection (10.04, df = 2, *p* = 0.007) expressed significant preference for foreign-made vaccines. The findings are summarised in Table 2.

### 3.4. COVID-19 Booster Vaccination Intent

Among the study participants, 1464 (71.1%) reported their intention for the COVID-19 booster dose uptake: 39.8% responded definitely yes and 31.3% responded probably yes. On the other hand, 17.4% of the respondents expressed that they were probably not intending to receive the COVID-19 booster dose uptake, and 11.5% also responded with definitely no intention for COVID-19 booster dose uptake. 

In the chi-square test, there was a significantly higher definite intention for COVID-19 booster dose uptake among males (39.84, df = 1, *p* < 0.001). There was also significant association between definite intention to receive a COVID-19 booster dose and nationality (10.80, df = 1, *p* = 0.001), marital status (8.13, df = 1, *p* = 0.017), education level (14.24, df = 1, *p* = 0.001), monthly income (24.25, df = 1, *p* < 0.001), and co-morbid medical illness (4.26, df = 1, *p* = 0.045). The findings are summarised in Table 3. In the independent sample student t-test, there was a significantly higher definite intention for receiving a COVID-19 booster dose among older individuals (*p* < 0.001). There was a significant association between the constructs in the HBM model and COVID-19 vaccine (booster) intention. The findings are summarised in Table 4.

In the binary logistic regression analysis, perception of the third (booster) dose of COVID-19 vaccine as a good idea to decrease fears regarding getting COVID-19 (OR = 2.28, CI 1.89–2.76) and perceptions that receiving the booster dose reduces the risk of getting COVID-19 or its complications (OR= 2.69, CI = 2.17–3.34), under the perceived benefit construct, were the strongest predictors of a definite intention to receive a COVID-19 booster dose. Being concerned with safety (OR= 0.40, CI 0.34–0.47) under the perceived barriers construct was the strongest significant correlate of having no definite intention to take the COVID-19 vaccination. The findings are summarised in Table 5. 

## 4. Discussion

This study found that overall, 71.1% of the study population reported their intention for COVID-19 vaccination booster dose uptake, and a definite intention was reported by 39.8%. The results of the present study are in agreement with similar published studies. A cross-sectional survey among HCWs from Czechia reported that 71.3% reported positive intention for a booster dose of COVID-19 vaccination uptake [20]. Another online survey among adults in Poland reported that 71% of the study population reported positive intention to take a booster dose of COVID-19 vaccination [21]. Another recent cross-sectional study among medical students from Japan reported similar positive intention among 84.5% of all the respondents [22]. A recent study from Algeria reported that 51.6% of the participants reported positive intention to take a booster dose of COVID-19 vaccination [23]. Another study among German university students and employees reported an 87.8% acceptance rate [24]. However, 11.5% of our study population also reported a definite intention not to receive the COVID-19 booster dose vaccination. A recent study exploring the intention for the uptake of the COVID-19 booster dose vaccine among 22,139 fully vaccinated adults in the UK found that around 4% of the study population reported unwillingness for COVID-19 booster dose vaccine uptake [17]. Another cross-sectional study among HCWs from Czechia reported that 12.2% of the study population reported hesitancy for the COVID-19 booster dose vaccination [20]. 

In addition, factors influencing the intention to receive a third (booster) dose of COVID-19 immunisation and the role of the HBM in predicting vaccination intention among KSA healthcare workers (HCW) were investigated in this study. The study was started before the Saudi government approved the COVID-19 booster dose vaccination for HCW. The findings of the present study showed that HBM constructs of perceived benefit and perceived barriers significantly predicted COVID-19 booster dose vaccination uptake. However, the other HBM constructs (susceptibility, severity, and cues to action) did not significantly influence the definite intention for the third (booster) dose of COVID-19 vaccination among HCWs in the KSA. To the best of our understanding, no published studies are available exploring COVID-19 booster uptake intention using HBM. However, previous studies that analysed COVID-19 vaccination intention using HBM also reported similar results. Similar to our study results, a Malaysian study reported that the HBM constructs (susceptibility, benefit, and barriers) significantly predicted a definite intention for COVID-19 vaccination uptake [25]. A recent Chinese study also reported the perceived benefit and the perceived severity constructs of HBM as significant predictors of a definite intention for COVID-19 vaccination uptake [26]. Another study from Saudi Arabia among the general population found the HBM constructs of perceived susceptibility and perceived benefit as significant predictors of a definite intention for COVID-19 vaccination uptake [10]. Based on these results, we argue that the promotion of educational activities among HCWs highlighting their vulnerabilities for COVID-19 infection along with providing scientific information regarding COVID-19 vaccines might significantly improve the third (booster) dose COVID-19 vaccination uptake among HCWs in the KSA. 

The study results showed that perceptions regarding the susceptibility to COVID-19 and worries regarding the likelihood of getting COVID-19 infection were less among the majority of the study population. One of the reasons for the lower level of perceived susceptibility among HCWs could be that the majority of them were fully vaccinated and the rate of COVID-19 infection in KSA at the time of the study was significantly less. However, lower perceived susceptibility for COVID-19 infection might lead to less compliance with preventive measures against COVID-19 infection [27]. There is a need to educate HCWs regarding their susceptivity towards COVID-19 infection, especially with the new variants such as Omicron, which carry the potential to infect even fully vaccinated individuals. However, a significant proportion of the study population perceived that COVID-19 infection can lead to severe complications and reported fear of COVID-19 infection. One of the reasons for this higher level of severity perception could be higher exposure of HCWs to various kinds of COVID-19-related complications during their clinical work. The majority of the study population shares the idea that the COVID-19 booster dose vaccination can reduce their susceptibility to COVID-19 infection and associated complications. The study results also showed that there are lower levels of barriers to the COVID-19 booster dose vaccination uptake. The study population perceived the booster dose strategy of COVID-19 vaccination to be safe and effective. Moreover, the majority of the study population also reported that their vaccination uptake behaviour will be influenced by the availability of adequate information regarding the COVID-19 booster dose vaccination and vaccination uptake by their colleagues. The overall study results regarding perceptions of HCWs regarding the third (booster) dose of COVID-19 vaccination is encouraging. However, there is a need to share more information regarding the COVID-19 booster dose vaccination strategy among HCWs in the KSA. Moreover, movements can request HCWs to publicly acknowledge their vaccination status with the COVID-19 booster dose vaccination so that it acts as a cue for other HCWs to receive the vaccination. 

This study’s results also showed that males, older participants, higher educational achievers, and respondents belonging to the high-income category had more acceptance for the COVID-19 booster dose vaccination, which is consistent with the published literature. Pal et al. explored attitudes towards the COVID-19 booster dose vaccine among US HCWs and found that younger age and lower level of educational attainment were associated with higher vaccine hesitancy [28]. Another similar study among HCWs in Czechia found that male gender and older age of the participants showed significantly higher acceptance for the COVID-19 booster dose vaccination [20]. Significantly higher definite intent for the COVID-19 booster dose vaccination uptake was also reported by respondents who were leading sedentary lifestyles and belonged to the non-Saudi population, probably due to their awareness of increased susceptibility for COVID-19 infection and its complications.

There are numerous limitations to this study that must be considered when evaluating its findings. The present study utilised an online survey methodology to collect data, and such study designs are prone to selection bias. The HCWs were inquired regarding the behavioural intention for COVID-19 booster dose vaccination uptake in this study, which may not reflect actual behaviour. Various confounding factors which might influence vaccination hesitancy such as co-morbid psychiatric symptoms were not included in this study. A number of factors that could predict hesitancy toward receiving vaccines, such as adverse events after previous vaccine administration, being pregnant, and hesitancy toward other types of vaccines, were not included in this study. Finally, the questionnaire did not investigate COVID-19 vaccine anamneses.

## 5. Conclusions

This study found that 71.1% of the study population reported their intention for COVID-19 booster dose vaccination uptake. Moreover, we also found that many HBM constructs predict vaccination intention for COVID-19 booster dose among HCWs in the KSA, which can be utilised to promote COVID-19 booster vaccination among HCWs. 

## Figures and Tables

**Figure 1 vaccines-10-00987-f001:**
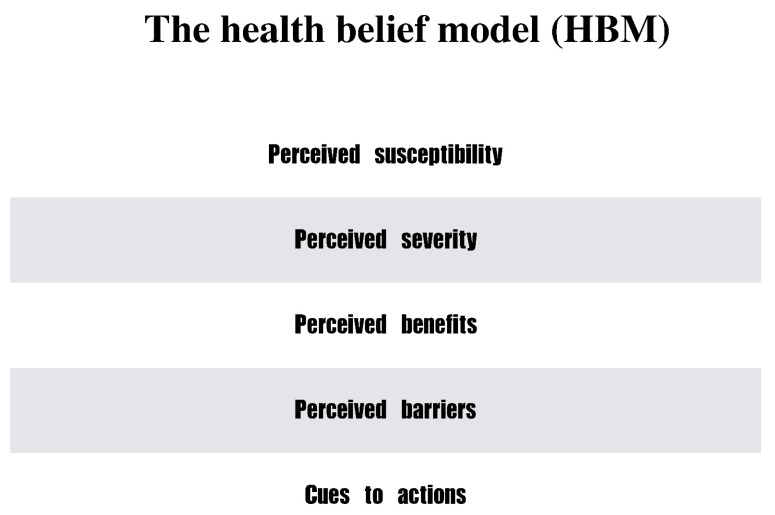
Constructs of health belief model (HBM).

**Table 1 vaccines-10-00987-t001:** Socio-demographic variables.

Variables		*n* (%)
Gender		
	Male	1023 (49.7)
	Female	1036 (50.3)
Nationality		
	Saudi	1743 (84.7)
	Non-Saudi	316 (15.3)
Education (*n* = 2031)		
	Bachelor	1415 (69.67)
	Masters/PhD	616 (30.33)
Marital status		
	Married	1074 (52.2)
	Widow/divorced/separated	68 (3.3)
	Single	917 (44.5)
Occupation (*n* = 2023)		
	Dentist	120 (5.93)
	Nurse	248 (12.26)
	Physician	767 (37.91)
	Pharmacist	149 (7.37)
	Technician	327 (16.16)
	Others	412 (20.37)
Provinces		
	Central	598 (29)
	Eastern	221 (10.7)
	Western	995 (48.3)
	Southern	162 (7.9)
	Northern	83 (4)
Monthly income (Saudi Arabian Riyal (SAR))		
	<5000	286 (13.9)
	5001–10,000	494 (24)
	10,001–15,000	499 (24.2)
	>15,000	780 (37.9)
Smoking status		
	Yes	383 (18.6)
	No	1481 (71.9)
	Ex-smoker	87 (4.2)
	Passive smoker	108 (5.2)
Physical activity		
	Sedentary	921 (44.7)
	<150 min/week	806 (39.1)
	>150 min/week	332 (16.1)
Managing COVID-19 patients		
	Yes	900 (43.7)
	No	1159 (56.3)
History of COVID-19 infection		
	Yes	485 (23.6)
	No	1574 (76.4)
Family member infected with COVID-19		
	Yes	1215 (59)
	No	844 (41)
Co-morbid illnesses		
	Yes	337 (16.4)
	No	1722 (83.6)
Rating of health status		
	Very good	1229 (59.7)
	Good	700 (34)
	Fair	120 (5.8)
	Poor	6 (0.3)
	Very poor	4 (0.2)

**Table 2 vaccines-10-00987-t002:** Association between vaccine preferences and socio-demographic variables.

Variable						
		Vaccine Preference				*p*-Value
		Participants	Local	Foreign-made	No preference	
Gender						
	Male	1023 (49.7)	167 (16.3)	378 (36.9)	478 (46.7)	<0.001
	Female	1036 (50.3)	126 (12.1)	309 (29.8)	601 (58)	
Nationality						
	Saudi	1743 (84.7)	267 (15.3)	549 (31.5)	927 (53.2)	<0.001
	Non-Saudi	316 (15.3)	26 (8.2)	138 (43.7)	152 (48.1)	
Education						
	High school	28 (1.3)	6 (21.4)	6 (21.4)	16 (57.1)	<0.001
	Bachelor	1415 (68.2)	228 (1.9)	418 (29.5)	769 (54.3)	
	Masters/PhD	616 (29.9)	59 (9.6)	263 (42.7)	294 (47.7)	
Monthly income						
	Less than 5000	286 (13.9)	47 (16.4)	72 (25.2)	167 (58.4)	<0.001
	5001–10,000	494 (24)	80 (16.2)	129 (26.1)	285 (57.7)	
	10,001–15,000	499 (24.2)	82 (16.4)	171 (34.3)	246 (49.3)	
	>15,000	780 (37.9)	84 (10.8)	315 (40.4)	381 (48.8)	
Location						<0.001
	Central	598 (29)	81 (13.5)	217 (36.3)	300 (50.2)	
	Eastern	221 (10.7)	33 (14.9)	76 (34.4)	112 (50.7)	
	Northern	83 (4)	19 (22.9)	23 (27.7)	41 (49.4)	
	Southern	162 (7.9)	44 (27.2)	34 (21)	84 (51.8)	
	Western	995 (48.3)	116 (11.6)	337 (33.9)	542 (54.5)	

**Table 3 vaccines-10-00987-t003:** Association between socio-demographic and clinical variables and COVID-19 vaccine third dose (booster) intention.

Variable				
		COVID-19 Vaccine (Booster) Intention		
		Definitely Yes	Probably Yes /Probably No/Definitely No	*p*-Value
**Gender**				
	Male	477 (46.6)	546 (53.4)	<0.001
	Female	342 (33)	694 (67)	
**Nationality**				
	Saudi	667 (38.3)	1076 (61.7)	0.001
	Non-Saudi	152 (48.1)	164 (51.9)	
**Marital status**				
	Married	445 (41.4)	629 (58.6)	0.017
	Widow/divorced/separated	35 (51.5)	33 (48.5)	
**Education**				
	High school and below	9 (32.1)	19 (67.9)	0.001
	Bachelor or diploma	527 (37.2)	888 (62.8)	
	Master and PhD	283 (45.9)	333 (54.1)	
**Monthly income**				
	<5000	102 (35.7)	184 (64.3)	<0.001
	5001–10,000	185 (37.4)	309 (62.6)	
	10,001–15,000	170 (34)	329 (66)	
	>15,000	362 (46.4)	418 (53.6)	
**Medical co-morbidities**				
	Yes	151 (44.8)	186 (55.2)	0.039
	No	668 (38.8)	1054 (61.2)	

**Table 4 vaccines-10-00987-t004:** Association between HBM variables and COVID-19 vaccine third dose (booster) intention.

Variable					
		Total Responses	COVID-19 Vaccine (Booster) Intention		
			Definitely Yes	Probably Yes /Probably No/Definitely No	*p*-Value
**Perceived Susceptibility of Contracting COVID-19**					
My chance of getting COVID-19 in the next few months is great					
	Strongly Agree	78 (3.8)	41 (52.6)	37 (47.4)	<0.001
	Agree	659 (32)	285 (43.2)	374 (56.8)	
	Disagree	1026 (49.8)	398 (38.8)	628 (61.2)	
	Strongly Disagree	294 (14.4)	95 (32.3)	201 (67.7)	
I am worried about the likelihood of getting COVID-19					
	Strongly Agree	157 (7.6)	66 (42)	91 (58)	<0.001
	Agree	672 (32.6)	307 (45.7)	365 (54.3)	
	Disagree	825 (40.1)	310 (37.6)	515 (62.4)	
	Strongly Disagree	405 (19.7)	136 (33.6)	269 (66.4)	
Getting COVID-19 is currently a possibility for me					
	Strongly Agree	260 (12.6)	125 (48)	135 (52)	<0.001
	Agree	1279 (62.1)	537 (42)	742 (58)	
	Disagree	380 (18.5)	125 (32.9)	255 (67.1)	
	Strongly Disagree	140 (6.8)	32 (22.8)	108 (77.1)	
**Perceived Severity**					
Complications from COVID-19 are serious					
	Strongly Agree	708 (34.4)	341 (48.2)	367 (51.8)	<0.001
	Agree	1026 (49.8)	389 (37.9)	637 (62.1)	
	Disagree	269 (13.1)	76 (28.3)	193 (71.7)	
	Strongly Disagree	56 (2.7)	13 (23.2)	43 (76.8)	
I will be very sick if I get COVID-19					
	Strongly Agree	96 (4.7)	44 (45.8)	52 (54.2)	0.580
	Agree	485 (23.6)	192 (39.6)	293 (60.4)	
	Disagree	1211 (58.8)	473 (39.1)	738 (60.9)	
	Strongly Disagree	267 (13)	110 (41.2)	157 (58.8)	
I am afraid of getting COVID-19					
	Strongly Agree	313 (15.2)	151 (48.2)	162 (51.8)	<0.001
	Agree	860 (41.8)	363 (42.2)	497 (57.8)	
	Disagree	628 (30.5)	225 (35.8)	403 (64.2)	
	Strongly Disagree	258 (12.5)	80 (31)	178 (69)	
**Perceived benefits of COVID-19 vaccination**					
Third (booster) dose of COVID-19 vaccine is a good idea because I feel less worried about catching COVID-19					
	Strongly Agree	305 (14.8)	248 (81.3)	57 (18.7)	<0.001
	Agree	821 (39.9)	432 (52.6)	389 (47.4)	
	Disagree	645 (31.3)	127 (19.7)	518 (80.3)	
	Strongly Disagree	288 (14)	12 (4.1)	276 (95.9)	
Receiving third (booster) dose of COVID-19 vaccine decreases my chance of getting COVID-19 or its complications					
	Strongly Agree	473 (23)	373 (78.9)	100 (21.1)	<0.001
	Agree	953 (46.3)	401 (42.1)	552 (57.9)	
	Disagree	422 (20.5)	42 (10)	380 (90)	
	Strongly Disagree	211 (10.2)	3 (1.4)	208 (98.6)	
**Perceived barriers of COVID-19 vaccination**					
Worry that possible side-effects of the third (booster) dose COVID-19 vaccine would interfere with my usual activities					
	Strongly Agree	284 (13.8)	48 (16.9)	236 (83.1)	<0.001
	Agree	665 (32.3)	189 (28.4)	476 (71.6)	
	Disagree	871 (42.3)	418 (48)	453 (52)	
	Strongly Disagree	239 (11.6)	164 (68.6)	75 (31.4)	
I am concerned about the efficacy of the third (booster) dose COVID-19 vaccine					
	Strongly Agree	301 (14.6)	36 (12)	265 (88)	<0.001
	Agree	587 (28.5)	126 (21.5)	461 (78.5)	
	Disagree	908 (44.1)	464 (51.1)	444 (48.9)	
	Strongly Disagree	263 (12.8)	193 (73.4)	70 (26.6)	
I am concerned about the safety of the third (booster) dose COVID-19 vaccine					
	Strongly Agree	275 (13.4)	24 (8.7)	251 (91.3)	<0.001
	Agree	504 (24.5)	86 (17.1)	418 (82.9)	
	Disagree	916 (44.5)	446 (48.7)	470 (51.3)	
	Strongly Disagree	364 (17.7)	263 (72.3)	101 (27.7)	
I am concerned of the faulty/fake COVID-19 vaccine					
	Strongly Agree	277 (13.5)	57 (20.6)	220 (79.4)	<0.001
	Agree	426 (20.7)	115 (27)	311 (73)	
	Disagree	803 (39)	331 (41.2)	472 (58.8)	
	Strongly Disagree	553 (26.9)	316 (57.1)	237 (42.9)	
**Cues to action**					
I will only take the third (booster) dose COVID-19 vaccine if I was given adequate information about it					
	Strongly Agree	616 (29.9)	361 (58.6)	255 (41.4)	<0.001
	Agree	936 (45.5)	364 (38.9)	572 (61.1)	
	Disagree	326 (15.8)	75 (23)	251 (77)	
	Strongly Disagree	181 (8.8)	19 (10.5)	162 (89.5)	
I will only take the third (booster) dose COVID-19 vaccine if the vaccine is taken by many healthcare workers in Saudi Arabia					
	Strongly Agree	444 (21.6)	250 (56.3)	194 (43.7)	<0.001
	Agree	831 (40.4)	319 (38.4)	512 (61.6)	
	Disagree	531 (25.8)	189 (35.6)	342 (64.4)	
	Strongly Disagree	253 (12.3)	61 (24.1)	192 (75.9)	

**Table 5 vaccines-10-00987-t005:** HBM predictors of definite COVID-19 vaccine third dose (booster) uptake (binary logistic regression analysis).

Variable	OR (Exp B)	Wald	df	*p*
**Constant**	0.401	8.445	1	0.004
Third (booster) dose of COVID-19 vaccine is a good idea because I feel less worried about catching COVID-19	2.28	71.85	1	<0.001
Receiving third (booster) dose of COVID-19 vaccine decreases my chance of getting COVID-19 or its complications	2.69	81.89	1	<0.001
I am concerned about the safety of the third (booster) dose COVID-19 vaccine	0.40	131.56	1	0.003

## Data Availability

The data presented in this study are available on request from the corresponding author. The data are not publicly available due to privacy concerns.

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
