# Peer review of "Predictors of the Third (Booster) Dose of COVID-19 Vaccine Intention among the Healthcare Workers in Saudi Arabia: An Online Cross-Sectional Survey"

_vaccines, 2022, doi:10.3390/vaccines10070987_

Round 1

Reviewer 1 Report

In this manuscript, authors conducted a survey to understand healthcare workers intention to receive third booster dose of COVID-19 in Saudi Arabia. This study provides timely information, conducted properly and written well. Suggestions to improve the manuscript are below:

1. In abstract and results, what are the values e.g., 8.13 on line 26. Is this and other values the chi-square value? Make it clearer. 

2. Tables need to be organized better by reducing the white (empty) spaces. 

3. What is the unit for monthly income? 

4. For tables 3, 4 and 5, provide percentages on parentheses. 

Author Response

Reviewer 1

In this manuscript, authors conducted a survey to understand healthcare workers intention to receive third booster dose of COVID-19 in Saudi Arabia. This study provides timely information, conducted properly and written well. Suggestions to improve the manuscript are below:

Answer: Thank you for the encouraging statements.

  1. In abstract and results, what are the values e.g., 8.13 on line 26. Is this and other values the chi-square value? Make it clearer. 

Answer: They were chi-square value. The values were removed as suggested by one of the reviewer. 

2. Tables need to be organized better by reducing the white (empty) spaces. 

Answer: The tables were edited accordingly

3. What is the unit for monthly income? 

Answer: It is Saudi Riyal

4. For tables 3, 4 and 5, provide percentages on parentheses. 

Answer: The tables were edited accordingly

Thank you

Reviewer 2 Report

1) It seems that the questionnaire did not investigate COVID vaccine anamneses; if this is the case, the authors should clearly state this in the methods and discuss this as an important limitation of the study. Previous studies aimed at evaluating the intention to receive booster dose (quoted by the authors) actually included only persons who were previously fully vaccinated, or, all the subjects with collection of the vaccine anamnesis. Since hesitancy to receive the third dose can be driven by the same factors that predict uncertainty to receive the first/second doses, the lack of vaccine anamnesis does not allow to disentangle the specific factors (if any) that can predict hesitancy to receive the third booster dose or any other doses.

2) A number of factors that could predict hesitancy toward receiving vaccines have been suggested from previus studies, f.i. adverse events after previous vaccine administration, being pregnant, hesitancy toward other types of vaccines... If these factors were not investigated, this should also be recognized as a study limitation.

3) It seems that age has not been analyzed as a potential predictor, or at least is not reported in the tables; it should be added (at least as a categorical variable, i.e. age classes), since several previous studies observed that younger age is associated with a lower vaccine uptake.

4) the reported associations between demographic characteristics and definite intention to receive booster dose (Table 4, text and abstract) actually do not allow to understand the direction of the associations; f.i. the definite intention is related to educational level, but in which direction? I suppose that the highest educational level was positively associated to definite intent: 46% (283/616) of Master/Phd expressed a definite intention, 32% (9/28) of those with High school or below.
The same applies for the associations reported in Table 5, text and abstract.
See also next comments 5 and 6.

5) The authors may perform a multivariable analysis (i.e. mutually adjusted) to evaluate potential demographic predictors of definitive COVID vaccine uptake, and also adjust the analyses for HBM predictors (Table 6) by socio-demographic factors.

6) Tables are very long and space-consuming; beside reducing line spaces, I think that Tables 1 and 2 could be omitted, by adding a column (titled: Participants, n(%)) to Tables 4 and 5.

7) Online surveys are prone to several biases, first of all selection bias, and this is correctly mentioned as a limitation of the study. The authors may try to estimate the magnitude of this bias (if any), as done in other studies quoted in the references.

Author Response

Reviwer 2                                                           

  1. It seems that the questionnaire did not investigate COVID vaccine anamneses; if this is the case, the authors should clearly state this in the methods and discuss this as an important limitation of the study. Previous studies aimed at evaluating the intention to receive booster dose (quoted by the authors) actually included only persons who were previously fully vaccinated, or, all the subjects with collection of the vaccine anamnesis. Since hesitancy to receive the third dose can be driven by the same factors that predict uncertainty to receive the first/second doses, the lack of vaccine anamnesis does not allow to disentangle the specific factors (if any) that can predict hesitancy to receive the third booster dose or any other doses.

Answer: We agree with reviewer that we haven’t systematically collected vaccine related informations from the participants, which is a major limitation of our study. We have added that as a major limitation of our study.

2) A number of factors that could predict hesitancy toward receiving vaccines have been suggested from previus studies, f.i. adverse events after previous vaccine administration, being pregnant, hesitancy toward other types of vaccines... If these factors were not investigated, this should also be recognized as a study limitation.

Answer: We agree with the reviewer. We added the following line in limitation; A number of factors that could predict hesitancy toward receiving vaccines such as adverse events after previous vaccine administration, being pregnant, and hesitancy toward other types of vaccines, were not included in this study. 

3) It seems that age has not been analyzed as a potential predictor, or at least is not reported in the tables; it should be added (at least as a categorical variable, i.e. age classes), since several previous studies observed that younger age is associated with a lower vaccine uptake.

Answer: We agree with the reviewer that age is an important predictor of vaccination intention. We have done independent sample student t-test and found a significantly higher definite intention for receiving COVID-19 booster dose among older individuals (p<0.001), which is consistent with the previous studies. Please see the result section under the subheading COVID-19 booster vaccination intent

4) the reported associations between demographic characteristics and definite intention to receive booster dose (Table 4, text and abstract) actually do not allow to understand the direction of the associations; f.i. the definite intention is related to educational level, but in which direction? I suppose that the highest educational level was positively associated to definite intent: 46% (283/616) of Master/Phd expressed a definite intention, 32% (9/28) of those with High school or below. 

The same applies for the associations reported in Table 5, text and abstract.

See also next comments 5 and 6.

Answer: We have edited the tables with percentages to make it more clearer.

5) The authors may perform a multivariable analysis (i.e. mutually adjusted) to evaluate potential demographic predictors of definitive COVID vaccine uptake, and also adjust the analyses for HBM predictors (Table 6) by socio-demographic factors.

Answer: Thank you for the suggestion. We would like to restrict our regression analysis with health belief model constructs only as it was the major objective of our study. 

6) Tables are very long and space-consuming; beside reducing line spaces, I think that Tables 1 and 2 could be omitted, by adding a column (titled: Participants, n(%)) to Tables 4 and 5.

Answer: Thank you for the suggestion. We kept the Table 1 to provide the all the frequencies of all the sociodemographic factors. 2nd table was removed.

7) Online surveys are prone to several biases, first of all selection bias, and this is correctly mentioned as a limitation of the study. The authors may try to estimate the magnitude of this bias (if any), as done in other studies quoted in the references.

Answer: We have added survey design as a major limitation of our study.

Thank you

Reviewer 3 Report

Dear authors,

The study conveys important findings on the prevalence and drivers of COVID-19 vaccine booster hesitancy (VBH) among Saudi healthcare workers (HCWs). The Introduction section is well written; however, it is a bit long. The methods and results are well described. The interpretation/discussion of findings was made appropriately. I have only a few points below that aim to help improve the quality of the manuscript.

1. The abstract is too long. Please make it shorter, especially the results part. There is no benefit from reporting degrees of freedom (df) or the chi-squared value here. Please report only the p-values.

2. Line 8: is still continuing? Please use the simple present tense.

3. The entire manuscript requires professional language proofreading/editing.

4. What do you mean by "mixed health belief"?

5. Whenever you report confidence intervals (CI), you should mention the cutoff. I assume it was 95% CI

6. Line 43: this sentence is wrong (or at least outdated). We have promising treatments now.

7. The aim and objectives of the study should be explicitly written at the end of the Introduction section.

8. Methods: split "Study Design and Participants" into three subsections.
- Study Design
- Study Setting
- Study Population / Participants

9. Line 111: Add the trademark (TM) sign after Google Forms, and follow the superscript rule.

10. How did you validate/test your instrument? Explain your validity and reliability testing in detail with values such as cronbach's alpha.

11. Please follow the STROBE guidelines for cross-sectional studies. Cite the STROBE guidelines and add the checklist as a supplementary file.

https://www.ncbi.nlm.nih.gov/pubmed/17938396

12. Please add one figure for the HBM with highlighting the constructs you have used and the items (questions).

13. Results: what was your response rate? please add one figure for the flowchart of your study.

13. Your discussion narrative may benefit from comparing your results to those found on COVID-19 VBH in other countries.

https://www.mdpi.com/2076-393X/10/4/621

http://journal.frontiersin.org/article/10.3389/fpubh.2022.846861/abstract

Overall the study was well conducted, and I believe that the points above can help improve the presentation of your methods & findings.

Sincerely,

Author Response

Reviewer 3

The study conveys important findings on the prevalence and drivers of COVID-19 vaccine booster hesitancy (VBH) among Saudi healthcare workers (HCWs). The Introduction section is well written; however, it is a bit long. The methods and results are well described. The interpretation/discussion of findings was made appropriately.

Answer: Thank you for your encouraging statements. 

I have only a few points below that aim to help improve the quality of the manuscript.

  1. The abstract is too long. Please make it shorter, especially the results part. There is no benefit from reporting degrees of freedom (df) or the chi-squared value here. Please report only the p-values.

Answer: The result part was edited as suggested. 

2. Line 8: is still continuing? Please use the simple present tense.

Answer: The sentence was edited as suggested.

3. The entire manuscript requires professional language proofreading/editing.

Answer: Thank you for the suggestion. The manuscript was edited for language.

4. What do you mean by "mixed health belief”?

Answer: By mixed health belief, we mean that participants carry conflicting beliefs regarding a health belief construct such as susceptibility to COVID-19.

5. Whenever you report confidence intervals (CI), you should mention the cutoff. I assume it was 95% CI

Answer: Yes, it is 95% CI. It was mentioned in the statistical analysis section.

6. Line 43: this sentence is wrong (or at least outdated). We have promising treatments now.

Answer: The sentence was edited as follows; The most important available option in the control of the pandemic at present is vaccination against COVID-19 infection

7. The aim and objectives of the study should be explicitly written at the end of the Introduction section.

Answer: Thank you for the suggestion. We have added the following line at the end of the introduction; The aim of the current study was to explore perceptions of the booster dose of COVID-19 vaccine among HCWs from KSA and the health belief predictors of intention to receive the booster dose of the vaccine. 

8. Methods: split "Study Design and Participants" into three subsections.

- Study Design

- Study Setting

  • Study Population / Participants

Answer: Thank you for the suggestion. Since it was an online survey study, we couldn’t have much to elaborate on study setting or participants. Hence, we retained the section as such. 

9. Line 111: Add the trademark (TM) sign after Google Forms, and follow the superscript rule.

Answer: The trademark sign was added. 

10. How did you validate/test your instrument? Explain your validity and reliability testing in detail with values such as cronbach's alpha.

Answer: We haven’t validated the the health belief model (HBM) instrument. We used the instrument based on a similar study from Saudi Arabia (Alobaidi, S. Predictors of Intent to Receive the COVID-19 Vaccination Among the Population in the Kingdom of Saudi Arabia: A Survey Study. J Multidiscip Healthc 2021, 14,1119-1128. doi: 10.2147/JMDH.S306654.). We haven’t validated the the health belief model (HBM) instrument separately for our study population. 

11. Please follow the STROBE guidelines for cross-sectional studies. Cite the STROBE guidelines and add the checklist as a supplementary file.

https://www.ncbi.nlm.nih.gov/pubmed/17938396

Answer: The STROBE guidelines for cross-sectional studies was cited in the manuscript as suggested. Please see reference number 18.

12. Please add one figure for the HBM with highlighting the constructs you have used and the items (questions).

Answer: Figure showing HBM constructs was added. We haven’t added the items as they are provided in the tables already.

13. Results: what was your response rate? please add one figure for the flowchart of your study.

Answer: As it was an online survey with snow ball camping technique, we can’t provide response rate and flowchart on response rate.

13. Your discussion narrative may benefit from comparing your results to those found on COVID-19 VBH in other countries.

https://www.mdpi.com/2076-393X/10/4/621

http://journal.frontiersin.org/article/10.3389/fpubh.2022.846861/abstract

Answer: Thank you for the suggestion. We have included both the studies in the discussion. Please see references 22 and 23.

Overall the study was well conducted, and I believe that the points above can help improve the presentation of your methods & findings.

Answer: Thank you for your constructive suggestions which have greatly improved our manuscript.

Thank you

Round 2

Reviewer 2 Report

Thanks for the acknowledgement of the limitations of the study.

Please uniform the style (character type and dimensions) of the tables.

Reviewer 3 Report

Dear authors,

Thank you for addressing my previous comments. I see everything now is good with the manuscript.

Only two minor issues:

- I am not sure if it should be written "Google TM forms".
I believe MDPI production team can advise better regarding this.

- It should be "95% CI" not only "CI". This is also a point that can be fixed during the production phase of the manuscript.

Sincerely,